# Role of the Host Genetic Susceptibility to 2009 Pandemic Influenza A H1N1

**DOI:** 10.3390/v13020344

**Published:** 2021-02-22

**Authors:** Gloria Pérez-Rubio, Marco Antonio Ponce-Gallegos, Bruno André Domínguez-Mazzocco, Jaime Ponce-Gallegos, Román Alejandro García-Ramírez, Ramcés Falfán-Valencia

**Affiliations:** 1HLA Laboratory, Instituto Nacional de Enfermedades Respiratorias Ismael Cosio Villegas, Mexico City 14080, Mexico; glofos@yahoo.com.mx (G.P.-R.); marcoapg@iner.gob.mx (M.A.P.-G.); bruno_andre_dm@hotmail.com (B.A.D.-M.); garra22noviembre@yahoo.com.mx (R.A.G.-R.); 2High Speciality Cardiology Unit “Korazón”, Puerta de Hierro Hospital, Tepic 63173, Nayarit, Mexico; jaimeponcegallegos@hotmail.com

**Keywords:** influenza, inflammation, genetic susceptibility, polymorphisms, cytokine storm

## Abstract

Influenza A virus (IAV) is the most common infectious agent in humans, and infects approximately 10–20% of the world’s population, resulting in 3–5 million hospitalizations per year. A scientific literature search was performed using the PubMed database and the Medical Subject Headings (MeSH) “Influenza A H1N1” and “Genetic susceptibility”. Due to the amount of information and evidence about genetic susceptibility generated from the studies carried out in the last influenza A H1N1 pandemic, studies published between January 2009 to May 2020 were considered; 119 papers were found. Several pathways are involved in the host defense against IAV infection (innate immune response, pro-inflammatory cytokines, chemokines, complement activation, and HLA molecules participating in viral antigen presentation). On the other hand, single nucleotide polymorphisms (SNPs) are a type of variation involving the change of a single base pair that can mean that encoded proteins do not carry out their functions properly, allowing higher viral replication and abnormal host response to infection, such as a cytokine storm. Some of the most studied SNPs associated with IAV infection genetic susceptibility are located in the *FCGR2A*, *C1QBP*, *CD55*, and *RPAIN* genes, affecting host immune responses through abnormal complement activation. Also, SNPs in *IFITM3* (which participates in endosomes and lysosomes fusion) represent some of the most critical polymorphisms associated with IAV infection, suggesting an ineffective virus clearance. Regarding inflammatory response genes, single nucleotide variants in *IL1B*, *TNF*, *LTA IL17A*, *IL8*, *IL6*, *IRAK2*, *PIK3CG*, and HLA complex are associated with altered phenotype in pro-inflammatory molecules, participating in IAV infection and the severest form of the disease.

## 1. Introduction

Influenza A virus (IAV), a single-stranded negative-sense RNA virus of the *Orthomyxoviridae* family, is the most common infectious agent in humans, causing significant morbidity and mortality in infants and the elderly every year [1,2]. Also, influenza infects approximately 10–20% of the world’s population resulting in 3–5 million hospitalizations each year and an estimated 87.1 billion dollars in total annual economic burden in the United States alone [3]. Since 1918, humankind has experienced three influenza pandemics: the “Asian” influenza pandemic, in 1957, the “Hong Kong” influenza pandemic in 1968, and the 2009 so-named “swine flu” pandemic. Although mild compared to that of 1918, these pandemics highlight the constant threat that the influenza virus poses to human health [4].

In April 2009, the first influenza A H1N1 cases were registered in Mexico and associated with a surprising number of deaths [5], and it spread rapidly throughout the world [6]. Worldwide, estimates of the crude hospitalization fatality risk (HFR), defined as the probability of death among H1N1 pdm09 cases which required hospitalization for medical reasons, ranged from 0% to 52%, with higher estimates from tertiary-care referral hospitals in countries with a lower gross domestic product. However, in wealthy countries, the estimation was 1%–3% in all settings [7].

## 2. Bibliometric Analysis

A scientific literature search was performed using the PubMed database; the Medical Subject Headings (MeSH) “Influenza A H1N1” AND “Genetic susceptibility” were used. Studies published between January 2009 to May 2020 were included due to the amount of information and evidence about genetic susceptibility generated from the studies carried out in the last influenza A H1N1 pandemic. The bibliometric analysis was performed using RISmed [8], and for the wordcloud creation with the keywords of articles included in this review, we used the wordcloud tool [9] in RStudio V 3.6.1 [10] following the workflow proposed by the libraries developer. Applying this criteria selection, 119 papers were identified. Figure 1 shows articles published with these MeSH by year in the period, as mentioned above, and journals where papers were published.

A wordcloud showing the main keywords terms in the bibliometric analysis is shown in Figure 2.

## 3. Inflammatory Response and IFITM3 Role in Influenza A Virus Infection

Through experimental and clinical studies, the systemic inflammatory dysregulation correlating with the disease’s severity and progression has been identified as one of the most important pathogenic mechanisms in infection [11,12]. Cytokine secretion by infected cells is necessary for the initiation of the immune response that controls virus replication [13]; also, immunopathological mechanisms such as hypercytokinemia (also known as cytokine storm) generally contribute to the more severe evolution of the infection [14,15].

A cytokine storm during viral infection is a prospective predictor of morbidity and mortality, yet the cellular sources remain undefined [16]. Critically ill patients who developed ARDS related to influenza A H1N1 virus infection had a slower decline in nasopharyngeal viral loads; had higher plasma levels of pro-inflammatory cytokines and chemokines; and were more likely to have bacterial coinfections, myocarditis, or viremia than patients in the survived-without-ARDS or the mild-disease groups. The hallmarks of the severity of disease were interleukin-(IL)-6, IL-8 and Tumor necrosis factor (TNF)-α [17]. The genes encoding these proteins are polymorphic, and specific alleles have been associated with susceptibility to different diseases, covering a broad spectrum of pathologies, from infectious to oncological, including pulmonary and systemic diseases [18,19,20].

These genes’ extensive polymorphism could be associated with the high mortality rate during the A H1N1 influenza pandemic in Mexico and its high prevalence. According to Borja et al. 2012, a fourth wave observed in Mexico during 2011–2012 was not reported in other countries. Therefore, these differences could be explained by patterns of genetic susceptibility in Mexicans or vaccination coverage failures [21].

Elevated serum levels of IL-1β and IL-6 have been identified as markers of severity in acute lung damage during influenza A H1N1 virus pdm09 infection; additionally, elevated levels of IL-1β are considered an early biomarker of the severity and progression of lung inflammation in patients who require mechanical ventilation and who do not respond to conventional antimicrobial treatments [22,23].

For example, the UK experienced two waves, one in spring–autumn 2009, followed by a more severe one in 2010–2011 [24,25], which was not seen in other European countries. These differences could be explained by geographic variations, previous immunity, control strategies, connectivity, public health, and for now, it remains a key for future research.

The defense mechanism that is provided by the innate immune system is the quintessential barrier, a specialized immune system in which different mucosa co-exist that fights the invasion of pathogens. The viral RNA is recognized as foreign by the different Pattern Recognition Receptors (PRRs) present in infected cells, which secrete type I interferons (IFNs), pro-inflammatory cytokines, eicosanoids, and chemokines [26]. Type I interferons, produced by macrophages, pneumocytes, dendritic cells, and plasmacytoid dendritic cells (pDCs), stimulate the expression of hundreds of collectively called IFN-stimulating genes (ISGs) in neighboring cells, which induce an antiviral state. Pro-inflammatory cytokines and eicosanoids lead to systemic and local inflammation, which results in fever and anorexia and instructs the adaptive system to respond to the influenza virus [27,28].

The innate immune response comprises a system of mobile lines that encode different receptors inspecting the intracellular and extracellular compartments for signs of infection and highly conserved microbial motifs, also called pathogen-associated molecular patterns (PAMPs), which are molecular signatures of pathogens that facilitate induction of the host immune response; PAMPs activate cellular PRRs such as toll-like receptors (TLRs) to induce immunity [29]. Vast arrays of pathogens enable PRRs in the absence of PRR-specific PAMPs. It is thought that, during infection, cellular factors can activate PRRs and thus indirectly fulfill the function of PAMPs [30]. PRRs are classified into several families. The Toll-like receptor (TLR) family consists of more than ten members, which enable innate immune cells to respond to a variety of PAMPs [31]. TLR3, TLR7, TLR8, and TLR9 represent a TLR subfamily that recognizes viral nucleic acid and can induce type I IFN [32]. More recently, it has become apparent that viral RNA is also detected by members of the RIG-I-like RNA helicase (RLH) family, such as RIG-I and Mda5 [31,32,33]. TLR and RLH differ in their cellular localization, ligand specificity, and downstream signaling pathways, suggesting that host cells have multiple defense mechanisms against viral infection. Among ISGs, 2′–5′-oligoadenylate synthetase (OAS) plays a critical role in antiviral immunity by synthesizing 2–5As, which induces RNA degradation by activating a latent RNase (RNase L) pathway [34].

After infection, viral components are recognized by several PRRs that promote downstream cellular and humoral responses, including the cytokine storm [35], a term used for the first time in 2003 to describe an immune response to influenza infection about influenza-associated encephalopathy [36]. The influenza-induced cytokine storm has been linked to aggressive pro-inflammatory responses and insufficient control of the anti-inflammatory responses [16]. Several experimental studies and clinical trials suggested that a cytokine storm correlates directly with widespread tissue damage and unfavorable prognosis of severe influenza [37]. Nevertheless, we have little understanding of the mechanisms that promote cytokine storms or why some individuals exhibit an excessive response to the virus that leads to hospitalization or death, while the majority of patients only develop a mild to moderate form of the disease, without this exaggerated inflammatory response.

Also, the innate immune system recognizes the influenza virus by members of some of the three different classes of PRRs. Increased cellular expression of TLR9, TLR8, TLR3, and TLR7 during influenza has been reported while TLR2 and TLR4 were suppressed [38]. TLR9 may be the critical receptor among pattern recognized receptors to recognize and bind to influenza A H1N1 virus [38].

The IL-1 receptor-associated kinases (IRAKs) are critical regulators of TLR/IL-1 signaling, critical regulators of mammalian inflammation, and innate immune response. The non-synonymous *IRAK2* variant rs708035 (coding D431E) increases NF-κB activity and leads to more NF-κB-dependent pro-inflammatory expression cytokines compared with IRAK2 wild type [39].

Genetic studies in mice have determined a specific role for each of the ISGs, including the antiviral myxovirus resistance protein 1 (MX1), the interferon-inducing transmembrane protein (IFITM), and PKR, in limiting virus infection and its spread [38].

The myxovirus resistance (MX) genes are evolutionarily conserved in nearly all vertebrates. MX gene expression is induced by interferons type I or III, and the corresponding gene products can inhibit a wide range of viruses [38]. The MX1 is an interferon-induced GTPase that plays an essential role in the mammalian cell defense against influenza A viruses. Mouse MX1 interacts with the influenza ribonucleoprotein complexes (vRNPs) and can prevent the interaction between polymerase basic 2 (PB2) and the nucleoprotein (NP) of influenza A viruses [40]. Human MxA can suppress the replication of Orthomyxoviridae viruses (influenza and Thogoto), rhabdovirus (vesicular stomatitis virus), and hepadnavirus (hepatitis B virus), and mouse MX1 inhibits influenza and Thogoto virus replication [41].

IFITMs are a family of small proteins that comprises five members, including immune-related IFITM1, 2, 3, 5, and 10, with IFITM3 being the most important in host defense against viral infections [42]. The human *IFITM* loci measure approximately 18kb long and are located on chromosome 11. All of the *IFITM* genes contain two exons in their structure. The interferon-induced transmembrane protein 3 (*IFITM3*) gene is an endogenous immune-related gene considered as a small ISGs, since it includes an interferon-stimulated response element (ISRE) in its promoter region, which promotes a robustly up-regulated expression of IFITM3 by the stimulus of all three types IFNs, resulting in a significant affinity of transcriptional factors, the most critical being *POLR2A*, *MYC*, *ELF1*, *PHF8*, *CHD1*, *TAF1*, *REST*, *SIN3AK20*, *SIN3A*, *IRF1*, *STAT1*, *TBP*, *STAT3*, *STAT2*, *ZBTB7A*, and *CTCF* [43,44,45,46].

IFITM3 is an essential antiviral factor that has been shown to restrict RNA viruses’ replication, including IAV, the West Nile virus, and Dengue virus [47]. The mechanism of how IFITM3 regulates viral replication is not entirely understood. However, recent evidence suggests that IFITM3 is part of the endosomal compartments, preventing viral entry into the cytoplasm, preventing virus membrane fusion with cells, and inhibiting the fusion of infected cells (syncytialization) [48,49,50]. Essentially, IFITM3-deficient mice are more susceptible to influenza virus infection [47].

In this sense, studies in the *IFITM3* knockout (KO) mouse model have demonstrated that gene suppression improves weight loss and mortality following influenza virus infections. Also, infected *IFITM3* KO mice developed several cardiac disorders, such as aberrant cardiac electrical activity, including decreased heart rate and irregular, arrhythmic RR (interbeat) intervals. In contrast, WT mice exhibited a mild decrease in heart rate without irregular RR intervals. Additionally, these mice were accompanied by increased activation of fibrotic pathways and fibrotic lesions in the heart, suggesting a protective role of IFITM3 in the heart [51,52].

## 4. Genetic Variants and Influenza A H1N1 Virus Infection

### 4.1. Polymorphisms in the Complement System and Antibodies-Related Genes

There is considerable variability in the disease severity resulting from infection with influenza viruses. There are primary determinants of this variability; among these are the virus’s intrinsic pathogenicity, acquired host factors (such as immunity and comorbidities), and inherent host susceptibility. Whereas the viral genetic determinants of influenza severity and host immunity have been intensively studied, host genetic determinants are much less well studied. In 2009, the World Health Organization identified studies of the role of host genetic factors on susceptibility to severe influenza as a priority [53,54].

Single nucleotide polymorphisms (SNPs) are genetic variants involving a particular base pair [55]. The genes coding for immune-response proteins are polymorphic, and specific alleles have been associated with susceptibility to respiratory diseases that cover a wide range of pathologies [18].

Reports have estimated an increase in the seasonal influenza-associated respiratory deaths each year worldwide, mostly affecting older individuals [56]. Promptly, an exploratory study was published, providing evidence that genetic factors played an essential role in determining the susceptibility of Mexican Mestizo individuals to the development of severe pneumonia in the first outbreak of influenza A H1N1 infection. The authors found significant associations of five SNPs (rs1801274, rs9856661, rs8070740, rs3786054, and rs3744714) located on chromosomes 1 and 3 with the development of severe pneumonia in patients with influenza A H1N1 virus infection [57]. Three of these SNPs occur in *FCGR2A* (Fc Fragment of IgG Receptor IIa) and *C1QBP* (Complement component 1 Q subcomponent-binding protein), genes that may affect host immune responses to and/or replicate the A H1N1 influenza virus [57]. Immune complexes and complement activation have been implicated in the pathogenesis of the severe disease. Also, severe illness was found to be associated with high titers of low-avidity, non-protective anti-influenza antibodies, leading to immune complex deposition and complement activation in the respiratory tract [58]. Interestingly, the *FCGR2A* gene affects immune complexes’ control, while *C1QBP* can activate the complement system.

The *FCGR2A* gene encodes the Fcγ receptor IIA (FcγRIIA), which binds immune complexes with high avidity [59]. The human *FCGR2A* gene in the 1q23 chromosome region encodes a member of the heterogeneous Fc fragment of the IgG receptor family of immune receptors. It contains a functional rs1801274 variation in exon 4, which leads to the amino acid alteration from histidine (H) to arginine (R) at position 131 of the FcγRIIA protein [60]. The homozygous His131 genotype (A/A) was significantly enriched in patients with severe pneumonia compared with healthy A H1N1 exposed household contacts who did not develop respiratory illness [57]. The His131 allele of *FCGR2A* (FcγRIIA-H131) has a higher affinity than the Arg131 allele (FcγRIIA-R131) for all human IgG subclasses. The affinity of FcγRIIA-R131 for IgG2 is notably reduced, and FcγRIIa-H131 is the only human Fcγ receptor that recognizes this IgG subclass efficiently [61]. Immunoglobulin engagement of activating-type Fc receptors such as FcγRIIA induces multiple pro-inflammatory events, including immune cell degranulation and transcriptional activation of cytokine-encoding genes. Some FcγRIIA alleles have been proven to modulate the phagocytes’ ability to bind/internalize IgG-opsonised particles, with FcγRIIA-H131 conferring higher phagocytic function [62].

On the other hand, SNPs in chromosome 17 have been associated with severe disease [57] and increased death risk [62]; the rs3786054, located in the *C1QBP* gene, encodes the protein gC1qR, which was initially identified as a high-affinity receptor for C1q [62]. C1q is the first subcomponent of the C1 complex of the classical pathway of complement activation [62], and gC1qR can activate this pathway [63]. gC1qR may also contribute to the activation of the classical pathway of Complement by the surface of activated platelets [63]. It suggests that the risk allele of *C1QBP* associated with severe A H1N1 disease is associated with increased complement activation [57].

### 4.2. CD55 and RPAIN Polymorphisms

In an initially small-scale genome-wide association study, with selection of Complement decay-accelerating factor (*CD55*) single-nucleotide polymorphisms in Chinese patients with severe to mild disease [64] and Sanger sequencing, the primary outcome analyzed was death [65]. The rs2564978 genotype TT carriers were significantly associated with a severe infection under a recessive model, after adjustment for clinical confounders [66] and hospitalization requirement [65]. Interestingly, in influenza A H1N1 pdm09 patients from Northern Greece, the rs2564978 TT genotype was associated with increased death risk too [62], and in Spain it was indirectly associated with influenza severity [67].

CD55 is a membrane-associated protein regulating complement activation by interfering with C3/C5 convertases both in the classical and alternative pathways and naturally protects host cells from pathogens’ damage. An allele-specific effect on CD55 expression was revealed and ascribed to a promoter indel variation in high linkage disequilibrium with rs2564978. The promoter variant with deletion exhibited significantly lower transcriptional activity. Also, the authors demonstrated that CD55 could protect respiratory epithelial cells from complement activation [66].

On the other hand, using a microarray with more than 50,000 genetic variants, Zúñiga et al. [57] found that *RPAIN* (Replication protein-A-interacting protein) gene, also known as *hRIP* (human Rev-interacting protein), located in chromosome 17p13, is associated with influenza A H1N1 virus infection severity in a Mexican mestizo population. The authors hypothesize that the risk allele of hRIP/RPAIN (rs8070740) associated with severe A H1N1 disease is associated with increased influenza replication because it has been described that this protein interacts strongly with nuclear export protein (NEP), transferring the influenza RNAs from the nucleus of infected cells to the cytoplasm [64,68,69].

### 4.3. Genetic Variants in IFITM3 and Influenza A H1N1

Several groups have investigated polymorphisms in the *IFITM3* gene for association with IAV infection, disease severity, and clinical characteristics. Some genetic variations affecting the IFITM expression or function might contribute to viral pathogenesis [43].

Two SNPs (rs34481144 and rs12252) in *IFITM3* have been widely studied and represent some of the most critical polymorphisms associated with IAV infection [70]. The rs34481144 A allele (located in the promoter region) enhances the binding of CTCF to the *IFITM3* promoter, leading to a repressive effect on *IFITM3* expression, causing a reduction in protein levels in endosomes and lysosomes. This allele has been associated with IAV infection in three different cohorts; in the FLU09 Cohort (participants met the clinical case definition of influenza virus infection at the enrollment time, or were asymptomatic household contacts of a participant with confirmed influenza infection), it was reported that there was a higher frequency of homozygosity of risk A allele in patients with severe illness than the mild cases, as well as an increased frequency of the A allele in the Cohort of Genentech challenge study (healthy volunteers between 18 to 45 years old and seronegative in hemagglutination inhibition assays against A H3N2 IAV) and the PICFlu cohort (a multicenter study of influenza critical illness in North American children admitted across 31 pediatric intensive care units) [46].

Conversely, David and co-workers [71] found a statistically significant protective effect of the rs34481144-A allele against severe influenza under the dominant model (OR = 0.26; 95% CI 0.07–0.97), which could be due to the minor allele’s reduced frequency and the small sample size employed.

On the other hand, the most studied SNP associated with severe outcomes of IAV infection is rs12252, which is a non-synonymous variation in the first exon of *IFITM3*. The substitution of the T common allele for the alternative C allele might create an alternative splicing site and generate an N-terminal truncated variant of *IFITM3* with 21 amino acid residues deletion [72]. The altered protein might be mostly translocated to the plasma membrane. Therefore, it cannot restrict viral infection by IAV and its full-length counterpart [50,73]. Figure 3 shows the proposed mechanism to explain how *IFITM3* rs12252 participates in IAV infection.

This SNP has been associated with severe IAV infection in different populations. For example, Everitt et al. [72] reported that in diverse Caucasian populations, the C allele of the rs12252 has a higher frequency in hospitalized patients with IAV infection than healthy controls. Besides, the CC genotype is found in ~70% of Chinese Han patients with severe IAV infection, compared with 25% in moderate disease [74]. Conversely, López-Rodríguez et al. [75] did not find any association with either alleles or genotypes of rs12252 and IAV infection in critically ill Spanish individuals and those with mild disease. Also, David et al. [71] did not find an association between rs12252/C allele with mild or severe IAV infection in a Portuguese population. Mills et al. [76] reported an association with the CC genotype of rs12252 and mild IAV infection, contrasting with previous studies where the association was founded with severely ill patients. These results could be related to a decreased C allele frequency in European populations compared with Asian populations.

Wellington and co-workers [70] described that according to the 1000 genomes project [76], there are remarkable differences in rs34481144 and rs12252 in different populations, described in detail in Table 1.

Interestingly, minor allele frequency (MAF) of rs34481144 in the European population is higher than Asian populations, while in the opposite, MAF of rs12252 is much higher in Asian than European populations. Also, American and African populations are in the middle between European and Asian populations’ frequency.

In addition, David et al. [71] also described discrete difference in genotype frequencies from Central Africa (largely Angola, GG: 82%, GA: 18%) and Portugal (GG: 45%, GA: 44%, AA: 11%), compared with African (GG: 90%, GA: 10%) and European (GG: 29%, GA: 49%, AA: 22%) references from 1000 genomes. López-Jiménez et al. [77] described for the first time the allele and genotype frequencies of rs12252 from Western Mexico (four states: Nayarit, Jalisco, Colima, and Michoacán), where the frequencies, in general, were T: 82% and C: 18% for alleles, and TT: 67%, TC: 30%, CC: 3%, which are very similar to the 1000 genomes project.

### 4.4. Inflammatory Response Genes and Influenza A H1N1

Since the end of the 20th century, it has been described that variability in cytokine production between individuals may be due in part to genetic factors, especially the presence of polymorphisms in important regulatory regions, such as promoters [12,13,78]. It has been determined that the most important pathological mechanism in influenza A H1N1 infection is the systemic dysregulation of the inflammatory response, which is correlated with the illness severity and progression [79,80,81,82]. Besides, immunopathological mechanisms, such as hypercytokinemia, contribute to the severest evolution of the IAV infection [83,84,85]. The playing role of the polymorphisms of the genes encoding these cytokines in the disease’s severity is not fully understood.

Cytokine production varies among individuals due to genetic factors, particularly polymorphisms in important regulatory regions such as promoters [12,13,86]. The role of polymorphisms in coding genes play in disease susceptibility and severity will be discussed.

Although it has been reported that some genes involved in inflammation are associated with respiratory diseases [87,88,89,90,91,92], investigations regarding genetic factors involved in the susceptibility and severity are scanty. Two primary examples in inflammatory response genes, where genetic variability is associated with an altered phenotype, are TNF-α and IL-1β; both are pleiotropic modulators critical in regulating inflammation.

Since the initial discovery by Wilson et al. more than 20 years ago, using reporter genes under the control of the two allele *TNF* promoters, it has been shown that TNF2 (*TNF*-238/A allele) is a much more potent transcriptional activator than the TNF1 common allele (*TNF*-238/G allele) in a human B cell line [78], polymorphisms in the *TNF* promoter region have been the subject of multiples studies. Genetic variations in the *TNF* promoter region have been associated with a range of autoimmune [93,94,95,96], infectious [97,98,99], and oncological diseases [100,101,102].

In a case-control analysis in a Mexican mestizo population, the *TNF* rs361525 (AA genotype), rs1800750 (AA), and *LTA* (Lymphotoxin alpha) rs909253 (AG) were associated with a higher risk of infection by pandemic influenza A H1N1 [103,104]. Although mortality of the A H1N1 patients was higher than that of the influenza-like illness (ILI) patients, only *LTA* rs909253 AG genotype showed a limited statistically significant association with mortality, suggesting that being a carrier of heterozygous rs909253 genotype in the *LTA* gene entails a poorer prognosis for this illness. Also, *TNF* rs1800629 GA and rs1800750 AA were associated with the severity of the clinical behavior. The first study demonstrated that the polymorphisms in genes related to the inflammatory response could be influencing the risk of infection and death by influenza A H1N1 virus [103]. A possible mechanism may be to form linkage disequilibrium between alleles, creating haplotypes that differentially affect cytokines and chemokines’ expression and activity, thereby resulting in a severer clinical course for the infection. Late, in the same population, the *TNF*-238 (rs361525) GA genotype was associated with an increased risk of disease severity [104]. In contrast, the *TNF*-308 (rs1800629) AA genotype was associated with influenza A H1N1 infection in an Egyptian population [105], and the G allele with susceptibility to severe disease in another Mexican population [23].

Another well studied pro-inflammatory protein/gene is IL-1β/*IL1B*; several studies have provided evidence about the association of *IL1B*-511 (rs16944) with gastric cancer [106], gastritis risk [107], including meta-analysis [108].

Studies in the Mexican population found the *IL1B* rs16944 AA genotype associated with a high amount of leukocytes [103]. In contrast, the rs3136558 CC genotype was associated with an increased risk, but rs16944 AG and rs3136558 TC were associated with a decreased risk of infection [104]. In an Iranian population, the rs16944 was associated with severe influenza disease [109]. In a Chinese population, the rs1143627 (*IL1B*) and rs17561 (*IL1A*) were found to be associated with susceptibility to A H1N1 pdm09 [110].

In humans, IL-1 exists in two forms, IL-1α (encoded by *IL1A* gene) and IL-1β (encoded by *IL1B* gene), both genes located on chromosome 2 [111,112]. IL-1α and IL-1β are inflammatory cytokines that play essential roles in recruiting the immune and inflammatory cells and developing adaptive immune responses [113]. In bronchoalveolar lavage fluids and lung homogenates, there is an early increase in IL-1 in temporal association with symptom presentation and lung pathology after infection with A/PR/8/34 H1N1 or A H1N1 [22,114]. The rs17561 is a non-synonymous variant (Ala114Ser) in IL-1α protein [115], suggesting that this genetic change may lead to a potential functional variation in host susceptibility to A H1N1 pdm09. It has also been related to high C-reactive protein levels, regulating the severe inflammatory response [116]. However, the exact mechanism needs to be further studied.

During IAV infection, the immune response is triggered by the influenza virus ion channel M2, an essential component for virus entry and replication, leading to the assembly of the inflammasome in macrophages and dendritic cells (DCs) [117,118]. Then, the inflammasome activation results in the cleavage of pro-IL-1β by caspase-1 and produces the mature form of IL-1β [119]. IL-1β may act with IL-6 to induce IFN-γ production by T cells [120] and promote RORγT expression and Th-17 polarization of CD4 T cells [121]. The Th-17 effector cells produced IL-17 and facilitated the recruitment of neutrophils and inflammation [121].

In this sense, the IL6 rs1818879 (GA) heterozygous genotype has been associated with severe influenza A H1N1 virus infection. Compared with ILIs, patients with severe pA H1N1 infections exhibit increased serum IL-5 and IL-6 levels [104].

According to a genetic association study, the *IL8* rs4073 AA genotype is considered a risk factor for influenza A H1N1 infected Egyptian patients [104], while in Mexican patients, it was associated with a higher value for partial arterial oxygen pressure (PaO_2_) mmHg (with PaO_2_ < 60 mmHg defined as a severe disease) [103]. Interestingly, the rs2275913 in the *IL17A* gene was associated with both risk of influenza A H1N1 infection and more severe disease [109]. In contrast, GG and AG genotypes were associated with seasonal influenza A/H3N2 risk of infection in the Iranian population [122].

As previously stated, the IAV causes a severe pulmonary disease characterized by intense leukocyte infiltration. Whereas the activation and recruitment of leukocytes are essential to control infection, excessive activation of neutrophils and macrophages might be harmful to the host [123,124]. Phosphoinositide-3 kinases (PI3Ks) are central signaling enzymes involved in cell growth, survival, and migration [125]. Class IB PI3K or phosphatidyl-inositol 3 kinase-gamma (PI3Kγ), mainly expressed by leukocytes, is involved in cell migration during inflammation. The SNPs rs17847825 and rs2230460 (A and T alleles, respectively) in the *PIK3CG* gene were significantly associated with protection from severe disease using the recessive model in patients infected with influenza A H1N1 pdm09 [126]. Figure 4 shows a graphical summary of genetic polymorphisms and their participation in IAV infection.

### 4.5. HLA System Genetic Variants and Influenza A H1N1

The human leukocyte antigen (HLA) super-locus is a genomic region in the chromosomal position 6p21 that encodes the six classical transplantation HLA genes and at least 132 protein-coding genes that have essential roles in the regulation of the immune system as well as some other fundamental molecular and cellular processes. This small segment of the human genome has been associated with more than 100 different diseases, including common diseases and various other autoimmune disorders [127]. The differential specificity, selectivity, and diversity of HLA directly reflects a fragile equilibrium between the interplay of molecular defense mechanisms against foreign antigens and autoimmunity acquired in the course of human evolution and migration. Selective combinations of HLA class I alleles exert different disease progression effects of infectious or autoimmune origin [128]. HLA class I is involved in both innate (NK cells) and cell-mediated (CD8+ cells) immune response, significantly contributing towards viral clearance and a decrease in the severity of influenza infection [129,130,131,132]. Cellular immune response to influenza viruses in adults is dominated by memory responses, as most individuals are repeatedly exposed to circulating influenza strains (either by natural infection or vaccination) throughout their lives. Seasonal changes in viral strains usually do not generate completely novel T-cell epitopes, so HLA class I or II-restricted epitopes from previous influenza seasons are present as identical or highly-homologous sequences in new seasonal influenza strains. However, in the case of a new H1N1 strain, influenza A H1N1 pdm09 was a reassortment of swine, human, and avian influenza A strains [133,134].

The HLA genetic association studies focusing on the role of susceptibility to influenza A H1N1 are exceedingly scanty. Only two researches have documented specific HLA alleles associated with influenza A H1N1 virus infection.

In a Mexican-mestizo population, a lower frequency of A*24:02:01 allele in patients compared to asymptomatic contacts was found [135], suggesting a possible protective effect. This result disaccords with a previous study where the A*24 serogroup (A24 family) correlates positively with severe A H1N1 infection and mortality [136]. Nevertheless, this information was acquired by employing database analysis of conserved proteomic regions-based experimental predictions of HLA binding affinity, showing that HLA alleles preferentially target conserved regions of viral proteins, phenomena known as HLA targeting efficiency [137]. The study did not consider HLA-restricted NK cell-mediated viral clearance, which may have introduced a certain degree of bias. The HLA-A*24 subtype is a potential ligand for KIR3DL1, rendering an inhibitory NK signal [138], probably reducing the immune system over-reactivity to influenza A H1N1 pdm09 virus, which is known to contribute to the significant pathobiology of the disease [139]. A*24:02 is in linkage disequilibrium with B*39:01 [140], which possesses high targeting efficiency scores. A*24 comprises a large portion of the world population; this allele family is more common in some indigenous groups and constitutes more than half of the global population, especially in Asian countries and several Native American populations [141]. There is variability in the A*24 frequencies in Mexican mestizos depending on the Amerindian contribution and the region studied [142]. In this study, only 8.33% of the patients had any allele of the A*24 serotype, whereas the contact subjects reach 18% (A*24:02:01) [135]. The overall allele frequency of A*24:02 in Mexican mestizos is 16.4% [142]. Recently, a cross-HLA allele T cell response against the influenza A virus peptides were detected among both HLA-A11(+) and HLA-A24(+) donors. Furthermore, cross-responses were found in the entire HLA-A3 supertype population (including HLA-A11, -A31, -A33, and -A30). The cross-allele antigenic peptides within the peptide pool were identified and characterized, and the crystal structures of the major histocompatibility complex (MHC)-peptide complexes were determined. The subsequent HLA-A24-defined cross-allele peptides recognized by the HLA-A11(+) population were shown to bind to the HLA-A*1101 molecule slightly [143]. In a preliminary study, the HLA-A*11 and HLA-B*35 alleles were found conferring susceptibility to influenza A (H1N1) in the Northeast India population pdm09 [144]. A*02 subtypes are considered a general protection factor against influenza A H1N1 infection [145].

The HLA dataset contains six A*68 alleles, and two belong to different supertypes: A*68:01 to the A3 supertype and A*68:02 to the A2 supertype. No association was found with A*68:02:01 to A H1N1/09 infection, while A*68:01:01, despite its low frequency (<2%), was only present in patients [135]. This is in concordance with the HLA targeting efficiency study results showing that A*68:01 subtypes correlate positively with A H1N1 pdm09 mortality rates but not A*68:02 [136]. The potential impairment of HLA-A*68:01-restricted CD8+ T cells to mount robust immune responses was recently investigated, demonstrating the immunodominance potential of influenza-specific CD8+ T cells presented by a risk HLA-A*68:01 molecule, and advocates for priming CD8+ T cell compartments in HLA-A*68:01-expressing individuals for establishment of pre-existing protective memory T cell pools.

Higher frequencies of B*39:06:02 and B*51:01:05 were observed in subjects suffering from influenza A H1N1. Despite high targeting efficiency scores, B*39:06:02 (Bw6) allele frequency was higher in A H1N1 patients than asymptomatic contacts, which could be due to selective pressure exerted by certain KIR-HLA combinations favoring under-reactivity of NK cells. Interestingly, the allele B*39:06 has been observed in several Amerindian populations with a frequency of around 5% [140] and possesses a high targeting efficiency score [136]. In contrast, B*39:01:01, another Bw6 supertype, showed a protective effect; this observation may be due to strong linkage disequilibrium with A*24:02 existing in Amerindian populations. A higher frequency of B*51:01:05 was also found in influenza A H1N1 patients. Another allele, HLA-B*51:01:01, an ancestral allele of Amerindian origin [140], had low frequency in A H1N1/09 patients, suggesting a possible protective role. In contrast with the HLA-A alleles, the HLA-B alleles bind more efficiently with the A H1N1 viral proteins’ conserved regions. Molecular subtypes of HLA-B*39 (B*39:01 and B*39:06), common in Amerindian populations, are associated with severe disease forms [136], which may be a result of these alleles in linkage disequilibrium with another region within or close to the HLA locus.

Despite its shallow frequency in the Mexican population, a high C*03:02:01 was observed in Mexican influenza A H1N1/09 patients [140]; however, this was not statistically significant after Bonferroni correction. On the other hand, C*03:03:01, C*03:04:01, and C*07:01:01 showed low frequency in the influenza patients group. Compared to HLA-A or HLA-B, HLA-C is less polymorphic and presents a more restricted repertoire of peptides and low cell surface expression [146,147,148]. A proportionally higher frequency of KIR2DL1 C2− C1+ and KIR2DL3 C1+ was reported in ICU A H1N1/09 patients (indigenous and nonindigenous), relative to healthy controls [139].

A relatively low frequency of the A*02:01:01-B*35:01:01-C*04:01:01 haplotype was observed among the A H1N1/09 patients. This specific HLA haplotype represents more than 2.5% in the control group and has been found only in Hispanic and Mexican populations [140,142] with the resolution employed in this study. An increase in the frequency of A*68:01:02-C*07:02:01 haplotype in the patients’ group was also observed, a relevant finding since linkage disequilibrium is very high in the region, rendering potential haplotypes and can potentially amplify the disease risk.

An analysis of the LIFT cohort [149] found that Indigenous Australians display a restricted and distinctive HLA profile confirming previous published serological studies [150]. Trough molecular HLA typing verified the predominant frequencies of HLA-A*02:01, 11:01, 24:02, 34:01 and HLA-B*13:01, 15:21, 40:01/02, 56:01/02. Such restriction in HLA diversity and HLA usage is likely to have arisen from an evolutionary bottleneck that established a small ancestral pool with limited HLA diversity. As HLA alleles’ variability evolves rapidly, it is intriguing that there is a high degree of conservation in Indigenous Australians [149]. This could be partly explained by limited mixing with other populations, long-term adaptation to local pathogens, and minimal exposure to new pathogens that might drive selection and/or the emergence of new variants. Before European contact in the eighteenth century, limited or no influenza exposure may explain a low prevalence of protective HLA variants for influenza [149]. Also, serologically-defined HLA-A2-homozygous lymphocytes, in contrast to heterozygous lymphocytes, did not synthesize detectable influenza virus RNA or protein after exposure to the virus. HLA-A2-homozygous lymphocytes (including both homozygous and heterozygous donors) did internalize infectious viruses but were not susceptible to lysis by autologous virus-specific cytotoxic T lymphocytes (“fratricide”). A similar intrinsic resistance to influenza virus infection was observed with HLA-A1- and HLA-A11-homozygous lymphocytes and HLA-B-homozygous lymphocytes, suggesting that a significant proportion of individuals within a population that is characterized by common expression of HLA class I alleles may possess lymphocytes that are not susceptible to influenza virus infection and thus to mutual virus-specific lysis [151].

Besides, it has been recently described that an SNP (rs2071888/G allele and GG genotype) in *TAPBP* (TAP binding protein or Tapasin), a critical cofactor required for the assembly of HLA class I with exogenous peptides obtained by intracellular degradation through proteasome, is associated with a higher risk for Influenza A H1N1 virus infection in a Mexican mestizo population, suggesting a critical role of the antigen presentation process in the development of the disease [152].

Finally, the lack of a universal vaccine against all serotypes of influenza A viruses and the recent progress on T cell-related vaccines against influenza A virus illustrate HLA-restricted cytotoxic T lymphocytes’ critical role in anti-influenza virus immunity. However, the diverse HLA alleles among humans complicates virus-specific cellular immunity research. Elucidation of cross-HLA allele T cell responses to influenza virus specificity requires further detailed work.

## 5. Conclusions

In recent years, several lines of evidence have determined that the role of host factors in genetic susceptibility to Influenza A Virus infection displays an essential part in the appropriate immune response to the virus, determining the outcome of infection. Numerous genes participate in diverse mechanisms against the viral response, such as pro-inflammatory pathways, complement activation, processing and antigen presentation, and intracellular control of viral replication; genetic variants in these genes could generate an abnormal function or decreased levels of the molecules, leading to higher viral replication and an exaggerated immune response to IAV infection. The evidence suggests that a variety of genetic variants can contribute to susceptibility to other influenza A strains (such as H3N2), not only A H1N1, since they share a similar structure, surface proteins, and genetic components that induce the same immune response. However, despite this amount of evidence, genetic susceptibility markers that are not linked to immunity have not been widely studied, probably because the efforts to study the molecular mechanisms are focused on elucidating the pathogenesis of IAV infection. So, this could be an essential field for future investigations.

Unfortunately, the evidence of the SNPs in the host genetic susceptibility to IAV is still inconsistent, and this can be partially explained by the differences in the genetic background of the populations around the world, especially in those that are considered as mestizo populations with a rich genetic variability, which is a product of the years of genetic recombination between ancestral Amerindian, Caucasian, African, and Asian populations. In this context, more collaborative research is required to provide a better understanding of the genetic determinants of the biological mechanisms of host susceptibility, which could result in early prevention, better diagnostic methods, and directed-therapy interventions at populations with a higher risk for developing a more severe form of the disease to offer a better prognosis in the future.

## Figures and Tables

**Figure 1 viruses-13-00344-f001:**
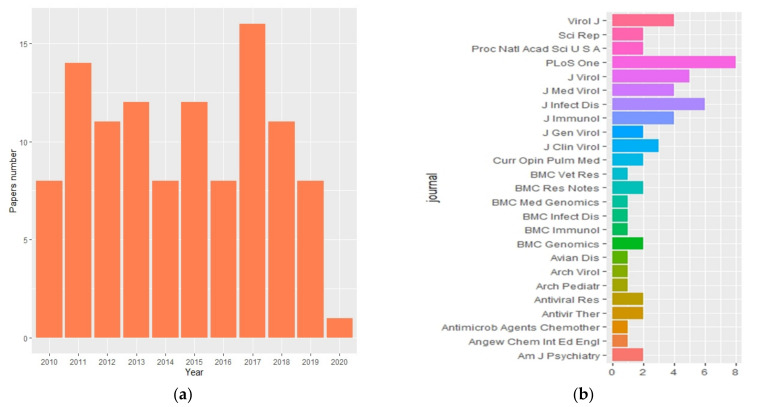
(**a**) Publications with the words “influenza A H1N1” and “genetic susceptibility” published between January 2009 to May 2020. (**b**) Top 25 journals where the 119 articles were published.

**Figure 2 viruses-13-00344-f002:**
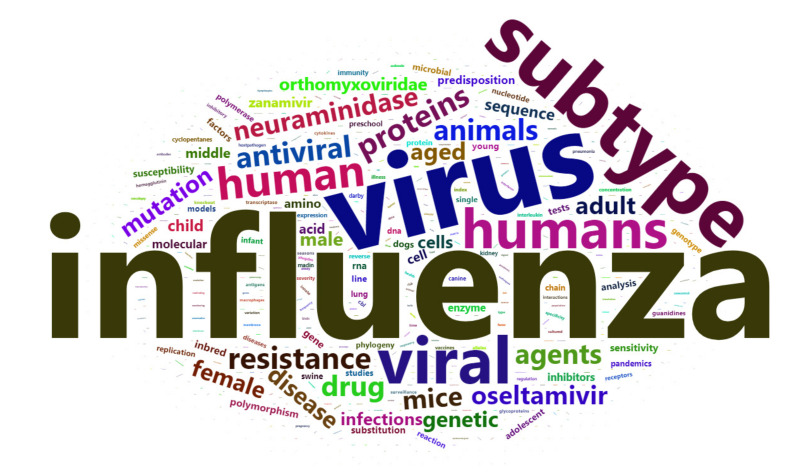
Depicting word cloud showing main keywords in the bibliometric analysis.

**Figure 3 viruses-13-00344-f003:**
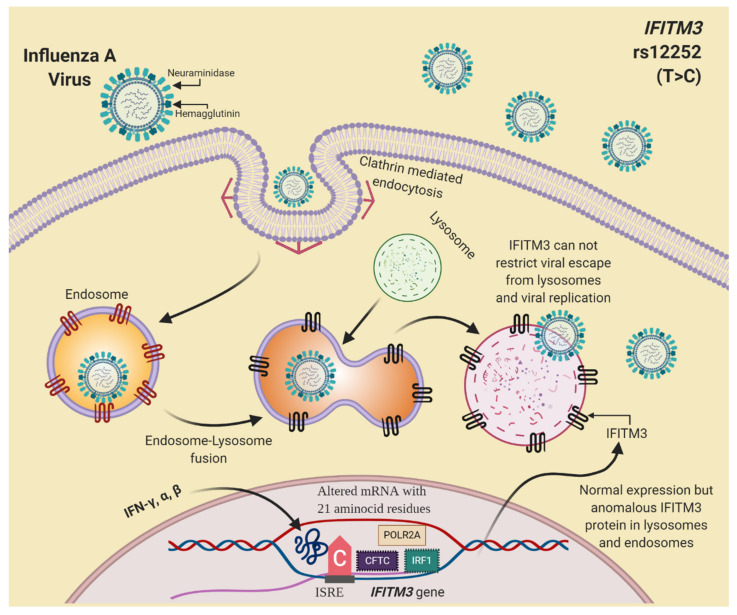
The proposed mechanism to explain *IFITM3* rs12252 participation in influenza A virus (IAV) infection. Created with BioRender.com.

**Figure 4 viruses-13-00344-f004:**
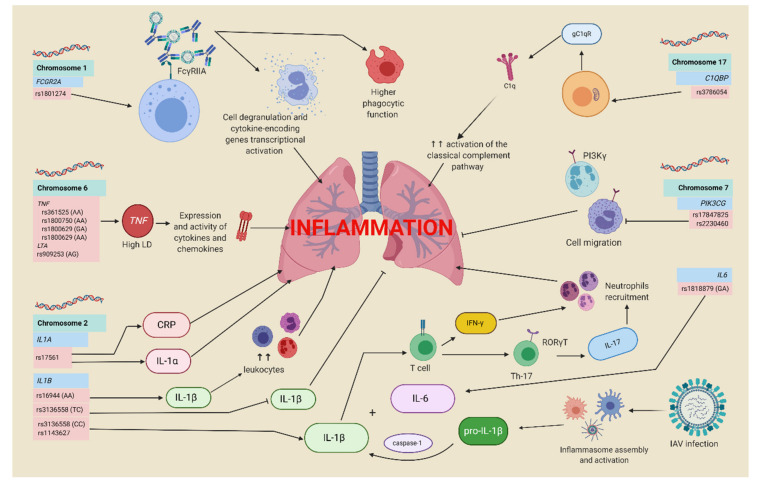
Graphical summary of genetic polymorphisms and their participation in IAV infection. Created with BioRender.com.

**Table 1 viruses-13-00344-t001:** Different *IFITM3* single nucleotide polymorphism (SNP) allele distribution by populations.

Population	rs34481144 G/A, (%)	rs12252 T/C, (%)
African	96/4	74/26
American	77/23	82/18
East Asian	99/1	47/53
European	54/46	96/4
South Asian	79/21	85/15
All	82/18	76/24

## Data Availability

No new data were created or analyzed in this study. Data sharing is not applicable in this article.

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
