# Peer review of "Role of the Host Genetic Susceptibility to 2009 Pandemic Influenza A H1N1"

_viruses, 2021, doi:10.3390/v13020344_

Round 1

Reviewer 1 Report

Perez-Rubio et al discussed about the host genetic susceptibility to influenza viruses infection. They complied total of 119 papers published between January 2009 to May 2020. They summarized the findings from these studies and categorize the reported host genes to the following groups: the genes involved in the innate immune responses and the genes involved in viral antigen presentation. Furthermore, they discussed the correlation between SNPs of those genes and the disease outcome of the influenza viral infection.

Major comments:
The authors have enlisted a detail experimental evidence from other studies to show that there is likely a correlation between the SNPs of these genes with severity of the disease. However, the most valuable point is to provide the expectation whether these SNPs could serve as an index to predict the outcome of disease. Based on the current information gathered by the authors, the answer is most likely no. In that case, the authors should give a detail discussion what is missing and why.

Specific points:
1. In figure 3, there is a typo in “IFITM3 ca not”

Author Response

Major comments:

The authors have enlisted a detail experimental evidence from other studies to show that there is likely a correlation between the SNPs of these genes with severity of the disease. However, the most valuable point is to provide the expectation whether these SNPs could serve as an index to predict the outcome of disease. Based on the current information gathered by the authors, the answer is most likely no. In that case, the authors should give a detail discussion what is missing and why.

  • Thank you for this interesting observation. We have included a brief paragraph in the Conclusion section where we have described what is missing and why the genetic susceptibility associated with IAV infection and the severe form is still unclear.

Specific points:

  1. In figure 3, there is a typo in “IFITM3 ca not”
  • Thank you. Now we have corrected the typo.

Reviewer 2 Report

This review summarizes the findings of 119 papers published in the last decade on genetic markers of susceptibility for severe influenza A virus infection, specifically with the 2009 pandemic H1N1 genotype. The paper is comprehensive and directly looked for all relevant publications rather than relying on the authors’ knowledge of the literature, which is commendable. My only concern is that I found the manuscript at times hard to follow, and it was not always clear where the authors were leading the reader. I have some comments aimed at improving readability and organization.

  • It would be useful if the authors stated in the abstract and the intro what their goal was with the literature research before diving into the results. Why they focus on H1N1 instead of all influenza A strains, for example. Also, although after thinking about it I realized why they picked the specific time period (they wanted to focus on the 2009 pandemic strain), it would be easier for the reader if it was explicitly stated.
  • The title should specify that they are only reviewing studies on the 2009 H1N1 pandemic strain, since this is their focus. That said, it would also be nice in the conclusion if they added some speculation as to whether they think the results would be the same with other influenza A strains.
  • Figure 2 does not really communicate much. None of the terms that are highlighted link to genetic susceptibility.
  • It is not entirely clear why there is a specific section (section #3) on IFITM3 (at least not until later in the review). Moreover, this section includes a paragraph on cytokine storm that is not clearly related to IFITM3 (lines 173-184). My suggestion would be to integrate current sections #3 and #4 into one and make sure it logically flows well.
  • There are several places (sections 5/6 for example) where the authors mention a gene and discuss some general information about susceptibility or link to influenza before they tell the reader what the gene is. For readability, they should tell the reader first what the genes are.
  • In section #5 RPAIN is mentioned but never discussed. What is this gene and how may it be connected to influenza infection?
  • Section #5 has a very generic title that could also encompass the material discussed in the next two sections. Maybe that should be changed to a more specific one that more closely defines the content of the section.
  • Could the authors also mention, very briefly towards the end, whether any genetic susceptibility markers have been found that are not clearly linked to immunity?

Language suggestions

Lines 9 and 32 – “infective pathogen” should be “infectious agent”

Lines 17- 19 – “Single nucleotide polymorphisms (SNPs) are a type of variation involving the change of a single base pair that can generate that encoded proteins do not carry out their functions properly” – the grammar seems a bit off. Maybe:  “Single nucleotide polymorphisms (SNPs) are a type of variation involving the change of a single base pair. SNPs can result in mutated proteins that do not carry out their functions properly” or something like it. That said, SNPs could also be in non-coding regions and change protein expression rather than function so maybe a more generic statement is necessary.

Line 42 – “spreads” should be “spread”

Line 68 – “The cytokines’ secretion” should be “Cytokine secretion”

Line 74 - “AH1N1” should be “A H1N1”

Line 95-97 – “For example, the UK … in other European countries.” – this sentence is not clear, because the authors mention “two waves” but then seem to talk about three (two in 2009 and one in 2010-2011). They should clarify this statement. Also, “after a more severe …” seems to imply the 2010-2011 wave was before the 2009 one, so it should probably be rephrased “followed by a more severe …”.

Lines 274 – “causing a minor presence of the protein” should probably be rephrased to something like “causing a reduction in protein levels”

Lines 276-279 – it would be useful if the author mentioned more about what the three cohorts are

Line 281 – what does “the dominant model of rs34481144” mean?

Lines 286-287 – “generate an N-terminal truncated variant of IFITM3 with 21 amino acid residues deletion [72], producing that the altered protein might be mostly translocated to the plasma membrane.” Should be changed to “generate an N-terminal truncated variant of IFITM3 with 21 amino acid residues deletion [72]. The altered protein might be mostly translocated to the plasma membrane.”

Line 346 – what is the gene LTA?

Line 358 – “Whereas” should be “In contrast”

Line 395 – define PaO2 mm Hg and explain what it implies about disease.

Lines 431-432 – The authors should clarify why that is an important piece of information about pH1N1.

Author Response

It would be useful if the authors stated in the abstract and the intro what their goal was with the literature research before diving into the results. Why they focus on H1N1 instead of all influenza A strains, for example. Also, although after thinking about it I realized why they picked the specific time period (they wanted to focus on the 2009 pandemic strain), it would be easier for the reader if it was explicitly stated.

  • Thank you for your observation. Now we have included a brief description of our principal aims in the abstract and bibliometric analysis sections.

The title should specify that they are only reviewing studies on the 2009 H1N1 pandemic strain, since this is their focus. That said, it would also be nice in the conclusion if they added some speculation as to whether they think the results would be the same with other influenza A strains.

  • Thank you. Now we have reformulated the title and conclusion.

Figure 2 does not really communicate much. None of the terms that are highlighted link to genetic susceptibility.

  • Thank you for your observation. You are right; most of the terms are not directly related to genetic susceptibility to influenza. However, all of these are the result of the bibliometric analysis, and at least three words are concerning the title and the aims of the manuscript (gene, polymorphism, genetic, susceptibility).

It is not entirely clear why there is a specific section (section #3) on IFITM3 (at least not until later in the review). Moreover, this section includes a paragraph on cytokine storm that is not clearly related to IFITM3 (lines 173-184). My suggestion would be to integrate current sections #3 and #4 into one and make sure it logically flows well.

  • Thank you, after your comment, we have integrated the two sections.

There are several places (sections 5/6 for example) where the authors mention a gene and discuss some general information about susceptibility or link to influenza before they tell the reader what the gene is. For readability, they should tell the reader first what the genes are.

  • Thank you. Now we have modified these concerns.

In section #5 RPAIN is mentioned but never discussed. What is this gene and how may it be connected to influenza infection?

  • Thank you. Now we have included a paragraph where we described the RPAIN role in influenza virus infection.

Section #5 has a very generic title that could also encompass the material discussed in the next two sections. Maybe that should be changed to a more specific one that more closely defines the content of the section.

  • Thank you for your observation. Now we have included sub-sections in section 4 to avoid these concerns.

Could the authors also mention, very briefly towards the end, whether any genetic susceptibility markers have been found that are not clearly linked to immunity?

  • This is an interesting observation. Now we have included a brief explanation in Conclusion section.

Language suggestions

Lines 9 and 32 – “infective pathogen” should be “infectious agent”

  • Now we have corrected these mistakes.

Lines 17- 19 – “Single nucleotide polymorphisms (SNPs) are a type of variation involving the change of a single base pair that can generate that encoded proteins do not carry out their functions properly” – the grammar seems a bit off. Maybe:  “Single nucleotide polymorphisms (SNPs) are a type of variation involving the change of a single base pair. SNPs can result in mutated proteins that do not carry out their functions properly” or something like it. That said, SNPs could also be in non-coding regions and change protein expression rather than function so maybe a more generic statement is necessary.

  • Thank you. Now we have changed the sentence.

Line 42 – “spreads” should be “spread”

  • Now we have corrected this mistake.

Line 68 – “The cytokines’ secretion” should be “Cytokine secretion”

  • Now we have corrected this mistake.

Line 74 - “AH1N1” should be “A H1N1”

  • Now we have corrected this mistake in the entire manuscript.

Line 95-97 – “For example, the UK … in other European countries.” – this sentence is not clear, because the authors mention “two waves” but then seem to talk about three (two in 2009 and one in 2010-2011). They should clarify this statement. Also, “after a more severe …” seems to imply the 2010-2011 wave was before the 2009 one, so it should probably be rephrased “followed by a more severe …”.

  • Thank you for your observation. Now we have corrected the sentence.

Lines 274 – “causing a minor presence of the protein” should probably be rephrased to something like “causing a reduction in protein levels”

  • Thank you for your observation. Now we have corrected the sentence.

Lines 276-279 – it would be useful if the author mentioned more about what the three cohorts are

  • Thank you. Now we have included a brief description about the three cohorts.

Line 281 – what does “the dominant model of rs34481144” mean?

  • Thank you for your question; single-nucleotide polymorphism consists of a major allele (M) and a minor allele (m). Thus, the genotype can be a major allele homozygote (MM), a heterozygote (Mm) or a minor allele homozygote (mm). A dominant model compares MM versus Mm + mm, and a recessive model compares MM + Mm versus mm (PMID: 26042205). In this particular case, results showed a statistical significant protective effect of the rs34481144-A allele against severe influenza under the dominant model (OR = 0.26; 95% CI 0.07–0.97). Now we have rephrased this sentence to better understanding it.

Lines 286-287 – “generate an N-terminal truncated variant of IFITM3 with 21 amino acid residues deletion [72], producing that the altered protein might be mostly translocated to the plasma membrane.” Should be changed to “generate an N-terminal truncated variant of IFITM3 with 21 amino acid residues deletion [72]. The altered protein might be mostly translocated to the plasma membrane.”

  • Thank you for your observation. Now we have changed those sentences.

Line 346 – what is the gene LTA?

  • Now we have clarified the LTA significate.

Line 358 – “Whereas” should be “In contrast”

  • Now we have changed the word.

Line 395 – define PaO2 mm Hg and explain what it implies about disease.

  • Now we have clarified the PaO2 significate and its implication.

Lines 431-432 – The authors should clarify why that is an important piece of information about pH1N1.

  • Thank you for your observation. To avoid later confusion, we have deleted these sentences.

Finally, I want to thank you for all your support and patience!

Round 2

Reviewer 1 Report

The authors have included information and made necessary modification in their new version.